# The COVID-19 Pandemic Increased Burnout and Bullying among Newly Graduated Nurses but Did Not Impact the Relationship between Burnout and Bullying and Self-Labelled Subjective Feeling of Being Bullied: A Cross-Sectional, Comparative Study

**DOI:** 10.3390/ijerph19031730

**Published:** 2022-02-02

**Authors:** Lena Serafin, Aleksandra Kusiak, Bożena Czarkowska-Pączek

**Affiliations:** Department of Clinical Nursing, Health Sciences Faculty, Medical University of Warsaw, 02-091 Warsaw, Poland; aleksandra@kusiaktrans.com (A.K.); bpaczek@wum.edu.pl (B.C.-P.)

**Keywords:** newly graduated nurses, bullying, burnout, COVID-19

## Abstract

(1) Background: The COVID-19 pandemic posed a great challenge to health care systems worldwide. Health care personnel, including nurses, work under high pressure and are overworked and overwhelmed, which results in a higher prevalence of burnout and workplace bullying, which further increases the intention to leave the nursing profession. (2) Methods: A comparative correlational and cross-sectional study design was adopted, and an online questionnaire was used to collect data between October 2019 and October 2021. Two hundred and fifty-seven newly graduated nurses participated in this study. The studied variable was measured using the Oldenburg Burnout Inventory, the Negative Acts Questionnaire, and metrics developed by the authors. (3) Results: The prevalence of bullying and burnout is significantly higher among nurses who worked during the COVID-19 pandemic than among those who worked before the pandemic, but the pandemic has not had an impact on the level of the subjective assessment of bullying. Working as a newly graduated nurse before or during the COVID-19 pandemic is a moderator between person-related bullying and its dimensions and disengagement. (4) Conclusions: Pandemics increase bullying and burnout among newly graduated nurses; however, the current challenges have caused some of this to remain unrevealed, the repercussions of which will appear with double strength later.

## 1. Introduction

The shortage of nurses that results from low nurse retention and instability in the nursing workforce due to increased turnover is currently a global concern [1]. Risk factors for the intention to leave the nursing profession have been identified, including workplace bullying and burnout syndrome, which are positively related to each other [2,3].

The current COVID-19 pandemic posed a great challenge to health care systems worldwide. Health care personnel, including nurses, work under high pressure and are overworked and overwhelmed, resulting in a higher prevalence of depressive symptoms, anxiety, and professional burnout or workplace bullying, which further increase the intent to leave the nursing profession [2,4,5]. Raso et al. [6] indicated that at least 11% of nurses intended to leave after the pandemic, while about 20% were undecided. Such a situation could intensify the instability in the nursing workforce. There is also evidence that more professionals retire than enter the profession every year, contributing to the nurse shortage and increasing the mean nurse age [7]. 

Burnout is defined as mental and physical exhaustion in response to prolonged emotional and interpersonal stressors in the workplace. In the 11th revision of the International Classification of Diseases (ICD-11), burnout is described as an “occupational phenomenon” and is recognised as a serious health problem [8]. Burnout was first described by psychologist Herbert Freudenberger in 1974, and pertains primarily to caregiving activities, including nursing [9,10]. Dall’Ora et al. identified mainly high workload, low staffing levels, long shifts, and low control as factors associated with burnout in nursing [11]. The potential consequences of burnout for patients and nurses are severe and might include, among others, reduced job performance resulting in poor quality of care and increased number of medical errors and adverse events, poor patient safety, lower job satisfaction, and an increased intention to leave [11]. 

The concept of workplace bullying was first introduced by Heinz Leymann, a Swedish psychiatrist, who used the term “mobbing” to describe a kind of workplace aggression towards employees [12]. Bullying includes any type of unethical behaviours, verbal, physical or emotional, directed by one or few individuals towards the victim, who is humiliated and disempowered due to it. Bullying is not clearly defined, and other terms, such as “psychological terror”, “psychological violence”, and “psychological harassment” are used to describe this workplace phenomenon as well [13,14]. According to the American Nurses Association Position Statement on Incivility, Bullying, and Workplace Violence, “bullying” regards harmful action taken by one perpetrator against the victim, while “mobbing” is rather the collective form of bullying [15]. According to Leymann, bullying is related to organisational factors, including deficiencies in work design, wrong leadership behaviour or low moral standards [12]. The outcomes of bullying are severe, voluminous, hazardous in terms of physical and mental health, and could result in an increased intention to leave the profession [2,5]. Investigations by Fino et al. revealed that verbal aggression might help to cope with adverse situationists by attenuated arousal in response to emotional stimuli. Additionally, female nurses, who constitute the majority of the nursing population, display cover forms of anger and aggression, which may have an impact on interpersonal relationships in the workplace and the development of burnout [16]. The available evidence highlights the many aspects of bullying and encourages further investigation into the phenomenon in the nursing work environment.

To ensure an adequate nursing workforce and diminish turnover, organisational attention to nurses’ well-being and a positive work environment is of the highest importance, especially in a pandemic [17]. Such activities should be based on documented evidence regarding these phenomena with a broader view of this issue in the context of newly graduated nurses (a nurses with less than 3 years of professional experience), who are especially vulnerable to pathological behaviours and a negative work environment [18]. According to Patricia Benner’s five-stages of nurses’ clinical competence theory, after two–three years of practice, nurses develop competency to see their actions in terms of long-range goals and gaining the Competent stage. At the first stage, named Novice, and at the second stage, named Advanced Beginner, they acquire new knowledge and skills very fast but still do not have enough in-depth experience [18].

Thus, considering the evidence mentioned above, important changes in the work environment of nurses caused by the COVID-19 pandemic may be a factor influencing the level of well-known and described problems in work psychology, such as bullying and burnout. The analysis of the long-term consequences of these phenomena, leaving the profession early, develops the need to analyse the issues among newly graduated nurses. Therefore, in this study, the following hypotheses were formulated: (1) Working during the COVID-19 pandemic increases the level of bullying among newly graduated nurses; (2) Working during the COVID-19 pandemic increases the level of burnout among newly graduated nurses; (3) There is a relationship between bulling and burnout among newly graduated nurses; (4) Working during the COVID-19 pandemic is a predictor of the relationship between bullying and burnout.

Therefore, this study aimed to analyse the occurrence and mutual relationship between bullying and burnout syndrome among newly graduated nurses. This aim is further defined by the moderating variable, namely the working time: before or during the pandemic.

## 2. Materials and Methods

### 2.1. Study Design

The comparative correlational and cross-sectional study design has been adopted to fulfil study objectives. Data were collected using an online questionnaire shared by social media, including specialised professional Facebook groups from October to December 2019 and from September 2020 to October 2021. Due to the announcement of the epidemic in Poland on 20th March 2020, the study group has been divided into two groups—respondents who participated in the study before the COVID-19 pandemic (from October to December 2019) were Group I respondents, and those who participated in the study during the COVID-19 pandemic (from September 2020 to October 2021) were Group II. Six months of pandemic professional experience has been used because the bullying questionnaire (Negative Acts Questionnaire) used in the present study asked about the past 6 months of experience [19].

The Strengthening the Reporting of Observational Studies in Epidemiology (STROBE) reporting guidelines were used in the framing and reporting. 

### 2.2. Sample and Setting

A convenience sample has been used in this study. Participants included Polish newly graduated nurses according to Benner’s theory [18]. Inclusion criteria were: working as a nurse in all settings performing direct patient care for not less than 6 months and not more than 36 months. Exclusion criteria included nurses not providing direct care to patients. 

The number of needed participants was calculated a priori using the G * Power 3.1 software, assuming the type of planned analysis with a d = 0.5, α = 0.05, and statistical power = 0.95. The minimum number of participants in each studied group was estimated to be at least 88.

A total of 257 nurses participated in this study. The final sample consisted of 212 participants who agreed to participate, returned a valid questionnaire, and fulfilled inclusion criteria. One hundred and twenty participants represented newly graduated nurses who worked before the COVID-19 pandemic (Group I), and 92 participants represented newly graduated nurses who worked during the COVID-19 pandemic (Group II).

### 2.3. Instrument

#### 2.3.1. General Sample Characteristic

Data on sociodemographic characteristics, such as age, sex, general seniority (in months), seniority in a current workplace (in months), education, and working setting (conservative, acute, intensive care, and other), were collected using a self-report questionnaire.

#### 2.3.2. Burnout

The participants’ burnout level was measured by using OLBI (Oldenburg Burnout Inventory), developed by Demerouti et al. (2003) in the Polish adaptation of Chirkowska-Smolak (2018) [20,21]. This scale assesses two aspects of burnout: disengagement, which refers to distancing oneself from one’s work in general, work object, and work content exhaustion defined as peoples’ intrinsic energetic resources, that is, emotional robustness, cognitive liveliness, and physical vigor. The scale consists of 16 statements that the respondent answered by marking which statement the respondent agreed with the most on a four-point Likert scale (1—”strongly agree”, 2—”agree”, 3—“disagree”, 4—“strongly disagree”). Eight statements are for disengagement and eight for exhaustion. The minimum possible number of points obtained in each of the tested dimensions is 8, and the maximum is 32. Burnout is determined when exhaustion was over 2.25 and disengagement was over 2.10, which correlate with physical symptoms [22]. Recent studies have used this interpretation to determine burnout prevalence [23,24].

In our study, the Cronbach’s Alpha for these two subscales was 0.79 for exhaustion and 0.67 for disengagement.

#### 2.3.3. Bullying

NAQ (Negative Acts Questionnaire), developed by Einarsen and Hoel (2001) in the Polish adaptation of Warszewska-Makuch (2007) [19,25], was used to measure bullying levels among newly graduated nurses. The NAQ includes 23 items related to negative acts that result in bullying. In 22 statements, participants were asked to specify the frequency of occurrence of negative acts using a five-point Likert scale (1—“never”, 2— “now and then”, 3—“monthly”, 4—“weekly” and 5—“daily”). The NAQ statements comprise three bullying dimensions: person-related (e.g., slander, social isolation and insinuation about someone’s mental health), work-related (e.g., giving a person too many, too few or too simple tasks, or persistently criticizing a person or their work), and intimidation-related (e.g., physical violence or the threat of violence). The questionnaire’s last (23) position includes a self-labelled definition of bullying based on participants subjectively judging whether they experienced bullying. We also used Leymann’s criteria for the prevalence of bullying assessment; respondents who experienced one or more negative acts, at least weekly and more often over 6 months, were classified as targets of bullying [26].

Cronbach’s alpha was previously reported to be 0.90 for the original version and 0.94 for the Polish version [19,25]. In our present study, Cronbach’s alpha was 0.93 for the total scale, 0.93 for person-related, 0.71 for work-related, and 0.73 for intimidation-related bullying.

### 2.4. Ethical Considerations

The Institutional Ethical Committee approved the authors’ affiliated university’s study (No. AKBE/205/2019), which was conducted following the Declaration of Helsinki. All participants were informed about the aim of the study and about the principles of voluntary participation and anonymity; returning the completed questionnaire was deemed to signify full acceptance of participation in the study.

### 2.5. Data Analysis

Statistical analyses were performed with the use of IBM SPSS Statistics 26.0. Descriptive statistics of the collective data were generated using standard parameters, including percentage, mean and standard deviation, median and range (minimum–maximum), skewness, and kurtosis. The Kolmogorov–Smirnov test was used to detect normal distribution. Between-group differences were analysed using the t-test for independent groups. The Spearman or Pearson’s test was used to analyse correlations between the variables. To compare the two groups in terms of occupational burnout and bullying prevalence, the analysis was performed using the χ2 test of independence or the Fisher’s exact test if the expected number was less than 5.

Then, using moderation analyses, we checked whether the time of commencement of a professional nursing career affected how the pandemic moderated the relationship between bullying and burnout syndrome. The analyses were performed using A. Hayes’ macro-PROCESS 4.0. For the analyses, α = 0.05 was assumed as the significance level.

## 3. Results

### 3.1. Characteristics of Investigated Group

In total, 212 newly graduated nurses participated in this study. One hundred and twenty of them represented Group I (nurses who worked before the COVID-19 pandemic), and 92 represented Group II (nurses who worked during the COVID-19 pandemic). The mean age of the studied group was 25 years old, and the mean period of seniority was 18.64 months (minimum–6 months; maximum–36 months). Most (67.5%, *N* = 143) of participants declared a bachelor’s degree in nursing. The newly graduated nurses in our study worked mainly at acute units and intensive care units: 34% ( *N* = 73) and 29.2% (*N* = 62), respectively. Detailed sociodemographic data are presented in Table 1.

### 3.2. Comparision of Bullying and Burnout between Newly Graduated Nurses who worked before COVID-19 Pandemic (Group I) and during COVID-19 Pandemic (Group II)

The analysis showed no statistically significant differences between the groups regarding bullying and all measured bullying dimensions, while significant differences were revealed for both dimensions of burnout. Newly graduated nurses who worked during the COVID-19 pandemic presented a higher level of exhaustion and disengagement than those who worked before the COVID-19 pandemic. The specific results of this analysis are presented in Table 2.

Comparison of bullying and burnout and its dimension prevalence based on the adopted threshold revealed that the percentage of exhaustion and disengagement in Group II was significantly higher than among Group I nurses, and the percentage of nurses without burnout was higher in Group I than among nurses in Group II. Moreover, in Group II, the percentage of nurses who met two burnout criteria was higher than in Group I (Table 3).

Comparing bullying prevalence based on the self-labelled subjective feeling of being bullied between two investigated groups revealed no statistically significant difference between nurses who worked before the COVID-19 pandemic and during the COVID-19 pandemic (Table 4).

### 3.3. Correlations between Bullying and Burnout in the Studied Group

To establish the relationship between the dimensions of bullying and the dimensions of burnout, an analysis of Pearson’s correlation was performed (for intimidation-related bullying, the Spearman correlation analysis was performed). The analysis showed positive correlations on a weak to moderate level, which means that the higher the level of perceived bullying on all dimensions, the higher the level of exhaustion and the higher the disengagement of the respondents (Table 5).

### 3.4. Working before the COVID-19 Pandemic and during the COVID-19 Pandemic as a Moderator of the Relationship between Mobbing and Burnout

To determine whether working before the COVID-19 pandemic and during the COVID-19 pandemic differentiated the relationship between bullying and its dimensions and the dimensions of burnout, moderation analysis was carried out using macro PROCESS 4.0. A. Hayes (model 1). A total of eight models were tested, four of which showed a significant role as a moderator. Working as a newly graduated nurse before or during the COVID-19 pandemic is a significant moderator of four relationships: person-related bullying and disengagement, work-related bullying and disengagement, intimidation-related bullying and disengagement, and bullying and disengagement. These models are presented below.

#### 3.4.1. Working as a Newly Graduated Nurse before or during the COVID-19 Pandemic as a Moderator between Person-Related Bullying and Disengagement

The first model considered working as a newly graduated nurse before or during the COVID-19 pandemic as a moderator between person-related bullying and disengagement. This model explained 11.6% of the variability of disengagement (increased by 4.0% after incorporating working as a newly graduated nurse before or during the COVID-19 pandemic into the model). The relationship between person-related bullying and disengagement was found to be significant for newly graduated nurses who working before COVID-19 pandemic (B = 1.53; SE = 0.36; *p* < 0.001; 95% CI (0.82; 2.24]). The higher the level of person-related bullying, the higher the disengagement. However, this relationship was insignificant among newly graduated nurses who working during COVID-19 pandemic (B = −0.31; SE = 0.48; *p* = 0.520; 95% CI [−1.27; 0.64]). The results are illustrated in Appendix A File S1.

#### 3.4.2. Working as a Newly Graduated Nurse before or during the COVID-19 Pandemic as a Moderator between Work-Related Bullying and Disengagement

The second model considered working as a newly graduated nurse before or during the COVID-19 pandemic as a moderator between work-related bullying and disengagement. This model explained 14.4% of the variability of disengagement (increased by 2.0% after incorporating working as a newly graduated nurse before or during the COVID-19 pandemic into the model). The relationship between work-related bullying and disengagement was found to be significant for newly graduated nurses who working before the COVID-19 pandemic (B = 1.93; SE = 0.38; *p* < 0.001; 95% CI (1.17; 2.68)). The higher the level of work-related bullying, the higher the disengagement. However, this relationship was insignificant among newly graduated nurses working during the COVID-19 pandemic (B = 0.46; SE = 0.53; *p* = 0.386; 95% CI [−0.59; 1.51]). The results are illustrated in Appendix A File S2.

#### 3.4.3. Working as a Newly Graduated Nurse before or during the COVID-19 Pandemic as a Moderator between Work-Related Bullying and Disengagement

The third model considered working as a newly graduated nurse before or during the COVID-19 pandemic as a moderator between intimidation-related bullying and disengagement. This model explained 10.9% of the variability of disengagement (increased by 2.7% after incorporating working as a newly graduated nurse before or during the COVID-19 pandemic into the model). The relationship between intimidation-related bullying and disengagement was found to be significant for newly graduated nurses who were working before the COVID-19 pandemic (B = 2.26; SE = 0.55; *p* = 0.001; 95% CI (1.17; 3.35)). The higher the level of intimidation-related bullying, the higher the disengagement. However, this relationship was insignificant among newly graduated nurses who were working during the COVID-19 pandemic (B = 0.02; SE = 0.70; *p* = 0.973; 95% CI([−1.36; 1.41)) (Appendix A File S3).

#### 3.4.4. Working as a newly graduated nurse before or during the COVID-19 pandemic as a moderator between bullying and disengagement

The fourth model considered working as a newly graduated nurse before or during the COVID-19 pandemic as a moderator between bullying and disengagement. This model explained 13.4% of the variability of disengagement (increased by 3.6% after incorporating working as a newly graduated nurse before or during the COVID-19 pandemic into the model). The relationship between bullying and disengagement was found to be significant for newly graduated nurses who working before the COVID-19 pandemic (B = 2.06; SE = 0.43; *p* < 0.001; 95% CI (1.22; 2.90)). The higher the level of bullying, the higher the disengagement. However, this relationship was insignificant among newly graduated nurses who were working during the COVID-19 pandemic (B = −0.14; SE = 0.61; *p* = 0.824; 95% CI (−1.34; 1.07)). The results are illustrated in Appendix A File S4.

## 4. Discussion

Due to the COVID-19 pandemic, nurses’ work schedules were severely disrupted. They faced several challenges, including deep stress due to the uncertainties of the disease progression, physical and emotional exhaustion, or concerns about direct exposure to COVID-19 at work. A rising number of nurses were infected with SARS-CoV-2 or died in the line of duty [27,28]. Additionally, many health care facilities experienced serious problems, such as poor planning, preparation, organisation, and leadership, and the failure to ensure adequate stocks of medical supplies, including personal protective equipment for medical staff [28,29,30]. The challenges resulting from the pandemic have exacerbated the already existing crisis in nursing worldwide and increased the possibility of phenomena related to occupational burnout or bullying. These issues especially refer to newly graduated nurses due to the shock resulting from the transition from nursing students to professional reality. Transition shock, which refers to the emotional burden that newly graduated nurses perceive due to the conflicting realities in clinical contexts, is positively and significantly related to turnover intention [31], while in contrast, a supportive work environment could be the key to retaining newly graduated nurses [32].

The literature also proved that workplace bullying is higher among nurses with less than 5 years of professional experience, and age and seniority are negatively correlated with workplace bullying [14,33,34]. The current data indicate that many nurses are quitting their jobs during pandemics or intend to leave, which could further amplify the mismatch between the supply and demand of nurses. According to the 2021 NSI Retention & RN Staffing Report [35], the turnover increased by 1.7% compared to last year, and 35.8% of hospitals reported a vacancy rate greater than 10%, compared to 23.7% in 2019.

The occurrence of burnout syndrome among nurses is present in all work settings. According to data from the meta-analysis of 113 studies, its overall prevalence is 11.3%, ranging from 0.2 to 47.83%, depending on the geographical region, the speciality of nurses, and the measurement tool used [8]. Burnout risk factors are identified, and they mostly refer to organisational issues, such as high demands, low job control, or high workload. However, they also include younger age and short professional experience, making newly graduated nurses especially vulnerable to this phenomenon [27,36,37]. The COVID-19 pandemic increased the prevalence of burnout syndrome among nurses since it intensified all previously identified risk factors and gave rise to new ones, such as increased risk for infection, decreased social support, insufficient material and human resources, and also the inability to have three regular meals, an adequate water supply, or sleep deprivation [27,30,38].

The prevalence of burnout syndrome before the pandemic in general, and in both evaluated dimensions in our study (disengagement and exhaustion) was almost twice as large as the highest reported in the meta-analysis mentioned above [8] and also reported by other researchers, and despite the high basic value, further increased during a pandemic [27,37]. However, individual works indicate that the prevalence of burnout syndrome before the pandemic in China was at a level comparable to our results [39]. The same refers to the prevalence of this phenomenon during pandemics [27,38]. It is difficult to explain the differences. Some of them could result from different measuring tools used in most studies, but we could hypothesise that most of all, they result from the fact that in our study, we investigated newly graduated nurses with little work experience during the transition from the study period to clinical practice, which is identified as the important risk factor for burnout syndrome [27]. However, some data indicate that the prevalence of burnout syndrome and low job satisfaction is higher among nurses with longer seniority and higher work experience [39]. Therefore, we cannot exclude the impact of organisational issues in the workplace, which worsened during the pandemic. The level of burnout syndrome identified by the total scores obtained in the questionnaire in our study was also high in both evaluated dimensions, and it also increased significantly during the COVID-19 pandemic, compared to the times before it. Other researchers using the same measurement tool also reported high but lower results rates in both dimensions among nurses in Singapore and Iran during a pandemic [40,41].

Zhang et al. [30] also identified a high level of burnout syndrome among Chinese nurses, despite different assessment methods being used. Given the different results on the prevalence and the level of burnout syndrome, especially during the COVID-19 pandemic, we conclude that burnout syndrome should be assessed in a specific workplace, and individual risk factors should be identified as the basis for planning individual and appropriate prevention with special attention to preparing nurses and also the organisation of health care facilities to better cope with following waves of COVID-19 or even the possibility of another pandemic in the future.

The level of burnout in both dimensions among the newly graduated nurses investigated in our study correlated with bullying, both self-labelled subjective feelings of being bullied, and all dimensions measured with the objective tool. This is in line with the results obtained by other researchers [33,42]. Bullying is a harmful experience leading to mental problems, from diminished self-esteem to suicide. It can also cause physical problems, including, among others, cardiovascular diseases, diabetes, fatigue, and pain. Bullying may increase the intent to leave the profession and threaten patient safety due to lower care quality [33,43]. There are limited data regarding the incidence of bullying during a pandemic. Most studies refer to its impact on the mental health of frontline care workers, including nurses. Asaoka et al. [44] showed that almost 10% of health care workers experienced workplace bullying during the COVID-19 pandemic; however, in this study, any health care professionals were included in the investigated group.

El Ghaziri et al. showed that 37.4% of 526 nurses from the investigated group experienced greater incivility at work during the COVID-19 pandemic than before, and 45.7% witnessed more incivility [45]. The prevalence of bullying in our study assessed using rigorous Leyman’s criteria was very high before the COVID-19 pandemic and increased significantly during the pandemic, reaching rates much higher than the literature data, which may be for the same reasons as indicated for burnout syndrome.

It is noteworthy that we did not confirm the impact of the COVID-19 pandemic on the level of bullying among newly graduated nurses, identified by the total scores obtained in the questionnaire, even though work stress is defined as the most important occupational risk factor and is strongly related to bullying [43]. We did not measure the perceived stress in our investigated group. However, it was proved that working as a nurse in a pandemic is associated with increased work stress [46]. We also did not confirm the impact of the COVID-19 pandemic on the self-labelled subjective feeling of being bullied. In this study, we examine the moderation role of the COVID-19 pandemic on the relationship between burnout syndrome and bullying. Our results show that the COVID-19 pandemic has removed this relationship, which was observed in the group working as a nurse before the COVID-19 pandemic. We could hypothesise that during the COVID-19 pandemic, there might be less opportunity to monitor bullying levels, and the victims may not pay so much attention to bullying because they recognise the higher priority of other goals related to saving people’s lives over their feelings. This could also result, at least in part, from the fact that the pandemic brought health care team members together [47]. However, considering the increase in the prevalence of bullying during the COVID-19 pandemic, such a situation requires measures to be taken to prevent bullying. Its effects may become evident after the end of the pandemic with double strength and much greater and more profound consequences.

Policies to prevent burnout and bullying should be implemented in every workplace. Monitoring these phenomena at the institutional level should be supplemented with the identification of the perpetrators, which would enable appropriate remedial action to be taken. The implementation of these activities seems to be crucial, especially when changes in the external environment create new and difficult challenges in the work of nurses. This can be severe for newly graduated nurses, who are focused on clinical competence in the first years of practice. The lack of a significant relationship between bullying and burnout among newly graduated nurses working during the COVID-19 pandemic, despite the earlier occurrence, shows how these external factors can cover significant problems in the work environment. Therefore, constant support for the ability to identify negative behaviours and the development of coping strategies should be included in transition programs. In the longer perspective, preventive action will increase the retention of nurses and decrease the turnover and intention to leave.

### Limitations

Our study is a cross-sectional study, which makes it difficult to establish a causal relationship between measured variables. Since the study was performed in Poland, cultural and contextual issues could impact the results. We used social media to recruit the responders, and people who used social media could differ in terms of technological access and resources. An addition to the analysis of control variables would allow for a more accurate depiction of the studied phenomenon.

## 5. Conclusions

Newly graduated nurses experience bullying and especially burnout syndrome at a high level. The COVID-19 pandemic worsened this situation; however, the current challenges caused some of this to stay unrevealed, yet the repercussions of this could appear with double strength in the future. A previously confirmed significant relationship between bullying and disengagement does not exist in a pandemic, which shows how external factors may cover important problems in the work environment. This highlights the need to constantly monitor the known phenomena and to consider new variables in order not to overlook the additional risk to nurses’ well-being.

A strong and adequate nursing workforce is essential for the health of every nation. Therefore, health care managers should constantly take appropriate action towards offering a better job environment and address solutions regarding another tough situation similar to the current pandemic that could happen in the future. Health care managers should implement appropriate interventions based on the current status of nurses’ characteristics and feelings to promote nurses’ health and well-being. Otherwise, the risk of nurses leaving the profession may increase, and the deficit of professionally active nurses will worsen.

## Figures and Tables

**Table 1 ijerph-19-01730-t001:** The characteristics of the investigated group.

	All (*N* = 212)	Group I (*N* = 120)	Group II (*N* = 92)
Sex, N *(%)*			
Female	203 (95.8)	116 (96.7)	87 (94.6)
Male	9 (4.2)	4 (3.3)	5 (5.4)
Age, *M (SD)*	25.53 (4.57)	25.66 (5.13)	25.37 (3.74)
Seniority (in months), *M (SD)*	18.64 (8.74)	18.34 (8.28)	19.02 (9.34)
Seniority in a current place (in months), *M (SD)*	15.78 (8.17)	15.66 (7.95)	15.93 (8.49)
Education, *N (%)*			
Bachelor’s degree	143 (67.5)	82 (68.3)	61 (66.3)
Master’s degree	68 (32.1)	37 (30.8)	31 (33.7)
Medical high school	1 (0.5)	1 (0.8)	0
Setting of work, *N (%)* Conservative unit	49 (23.1)	26 (21.7)	23 (25.0)
Acute unit	72 (34.0)	40 (33.3)	32 (34.8)
Intensive care unit	62 (29.2)	34 (28.3)	28 (30.4)
Other	29 (13.7)	20 (16.7)	9 (9.8)

*N*—numbers of respondents, M—mean, SD—standard deviation.

**Table 2 ijerph-19-01730-t002:** Comparison of bullying and burnout between Group I and Group II.

	Group I (*N* = 120)		Group II (*N* = 92)				95% *CI*		
	*M*	*SD*	*M*	*SD*	*t*	*p*-Value	*LL*	*UL*	Cohen’s d
Person-related bullying	2.08	0.96	2.23	0.87	−0.97	0.332	−0.36	0.12	0.16
Work-related bullying	2.31	0.88	2.47	0.72	−1.43	0.154	−0.38	0.06	0.20
Intimidation-related bullying	1.42	0.63	1.43	0.56	−0.15	0.879	−0.17	0.15	0.02
Bullying	1.99	0.80	2.09	0.64	−1.06	0.289	−0.30	0.09	0.14
Exhaustion	20.68	4.47	23.74	3.63	−5.35	<0.001	−4.18	−1.93	0.74
Disengagement	19.77	4.09	21.30	3.60	−2.83	0.005	−2.59	−0.46	0.39

M—mean; SD—standard deviation, t—t-statistic; LL—lower level; UL—upper level, Cohen’s d— effect size.

**Table 3 ijerph-19-01730-t003:** Relationship between burnout and its dimensions and bullying prevalence in Group I and Group II.

		Group I		Group II				
		*N*	*%*	*N*	*%*	χ^2^	*p*	*φ/V*
Bullying	Below threshold	59 _a_	38.3	20 _b_	20.2	9.20	0.002	0.19
	Above threshold	95 _a_	61.7	79 _b_	79.8			
Exhaustion	Below threshold	24 _a_	20.0	6 _b_	6,5	7.79	0.005	0.19
	Above threshold	96 _a_	80.0	86 _b_	93.5			
Disengagement	Below threshold	24 _a_	20.0	9 _b_	9.8	4.14	0.042	0.14
	Above threshold	96 _a_	80.0	83 _b_	90.2			
Burnout	No burnout	15 _a_	12.5	2 _b_	2.2	8.47	0.015	0.20
	One criterion met	18 _a_	15.0	11 _a_	12.0			
	Two criteria met	87 _a_	72.5	79 _b_	85.9			

*n*- number of participants, *%*—percentage, χ^2^—Chi square test, *p*—significance, *φ/V*—phi coefficient/Cramér’s V, _a_-the values in the columns that do not share the letter index differ at the level of p <0.05 (Bonferroni correction).

**Table 4 ijerph-19-01730-t004:** Comparison of bullying prevalence based on the self-labelled subjective feeling of being bullied between two investigated groups.

Have You Been Bullied?	Group I		Group II			
	*N*	*%*	*N*	*%*	*p*	*V*
Never	66	55.0	45	48.9	0.224	0.16
Now and then	23	19.2	24	26.1		
Monthly	17	14.2	18	19.6		
Weekly	7	5.8	4	4.3		
Daily	7	5.8	1	1.1		

*N*—number of participants, *%*—percentage, *p*—significance, *V*—Cramér’s V.

**Table 5 ijerph-19-01730-t005:** Correlations between bullying and burnout.

	Exhaustion		Disengagement	
	*r*	*p*	*r*	*p*
Person-related bullying	0.304	<0.001	0.211	0.002
Work-related bullying	0.365	<0.001	0.313	<0.001
Intimidation-related bullying *	0.267	<0.001	0.215	0.002
Bullying	0.346	<0.001	0.260	<0.001

*—Spearman’s rho, r—Pearson’s correlation, *p*—significance.

## Data Availability

Data are available upon reasonable request from the corresponding author.

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
