# Peer review of "The COVID-19 Pandemic Increased Burnout and Bullying among Newly Graduated Nurses but Did Not Impact the Relationship between Burnout and Bullying and Self-Labelled Subjective Feeling of Being Bullied: A Cross-Sectional, Comparative Study"

_ijerph, 2022, doi:10.3390/ijerph19031730_

Round 1

Reviewer 1 Report

  1. This manuscript aims to analyze the occurrence and mutual relationship between bullying and burnout syndrome among novice nurses. Even though this subject is very timely and relevant, my greatest concern is with the contribution of the present research. Authors should more explicitly lay out the importance of the research to the readers.
  2. The author should add a theoretical background to support the hypotheses. In line with this, I suggest that the hypotheses should be clearly formulated through the theoretical background.
  3. Please describe Benner’s theory explaining the choice.
  4. Did you use any control variables? If not, please explain why.
  5. What are the theoretical implications of the research?
  6. The implications for practice should be more developed and clearly stated.

Author Response

Thank you very much for your comments. They were very helpful for improving our manuscript. Based on the comments, we have made the following changes, which are summarized below.

Reviewer 2 Report

You try to develop a correlation between bullying and exhaustion.

May be the level of exhaustion is related to the hours working per day or per week.

I have still trouble understanding the bullying factor.

A detailed definition of what the various bullying means would be helpful.

What is the person-related bullying?

What is work-related Bullying?

What is Intimidation-related bullying?

The bullying directed at novice nurses initiates from managers, co-workers, patients, patient’s families, or others?

Without clear understanding of the source of the bullying, it really makes little sense discussing the results.

You would also think that the novice nurses who practiced before pandemic would be better equipped to handle the stress of the pandemic and burn out.

The conclusion is lacking a clear understanding of the findings of the study. All the hypotheses should be defined early on and results of the findings explained in the conclusion.

Author Response

(The authors gave the same response as above.)

Reviewer 3 Report

The present paper investigates the relationship between burnout and bullying among novice nurses in the healthcare setting during the COVID-19 pandemic as compared to prior to the pandemic. The paper is wellwritten and I find the issue quite relevant not only because nurses, amongst healthcare professionals, are consistently found to report the highest levels of pandemic related burnout, but also because this situation mirrors nurses’ professional life independently of a pandemic situation. Nursing is naturally an emotionally demanding profession for many reasons and nurses have the highest exposure to interpersonal aggression by patients, family members, visitors, fellow colleagues and other healthcare professionals. While it goes without saying that novice nurses are particularly vulnerable to the pressures and demands of nursing practice as they are to negative effects of aggression and bullying in the workplace, a less intuitive account is that the fact that aggression is so widespread may indicate that it is used as a defense mechanism to manage a wide range of adverse and emotionally charged situations. Indeed there are studies that suggest that at least verbal aggression ca have a facilitating role in the management of difficult work situations by novice nurses.  Far from justifying the presence and use of aggressive behaviors and language in the workplace by nurses, I suggest authors to consider this related literature (see for instance https://onlinelibrary.wiley.com/doi/abs/10.1111/jan.13936 )in the attempt to look at the phenomenon from a wider perspective. I believe that the introduction section and the overall quality of the paper could benefit from mentioning this research.  

Author Response

(The authors gave the same response as above.)
